# The Development of a Certification and Grading Procedure for German SCI Units

**DOI:** 10.3390/healthcare12242529

**Published:** 2024-12-13

**Authors:** Rainer Abel, Kerstin Rehahn

**Affiliations:** 1Department of Orthopedic Surgery, Klinikum Bayreuth GmbH, 95445 Bayreuth, Germany; 2Behandlungszentrum für Rückenmarkverletzte, BG Klinikum Unfallkrankenhaus Berlin, Warener Straße 7, 12683 Berlin, Germany; kerstin.rehahn@ukb.de

**Keywords:** spinal cord injury treatment, certification German, medico legal system

## Abstract

Background/Objectives: Spinal cord injury (SCI) care in Germany was established after the Second World War, following Guttman’s philosophy that post-SCI rehabilitation should not be separated from the acute treatment phase. Reimbursement is negotiated with only rudimentary eligibility requirements. Over time, however, more and more centers have emerged that offer “only” rehabilitation. Therefore, in 2014, the German-Speaking Paraplegic Society (DMGP) saw the need to establish a certification and grading process to protect existing centers and enable qualified reimbursement negotiations. Methods: In a modified delpi process, key data reflecting the human resources and equipment of the units were collected, and, after lengthy consensus negotiations, a grading was proposed which recognizes four levels of performance. Level Ia—24/7 coverage for all SCI-related emergencies (including intensive care unit care and surgery)—to Level IIb (rehab only, no intensive care unit). Results: In 2019, the grading was accepted by the extended board of the DMGP, and all but one of the 27 SCI centers applied to be graded and certified according to a self-reported questionnaire (2020). Conclusions: The development of the certification requirements and grading system was a complicated process, but it was possible to reach a solution which allowed its acceptance by all German SCI centers. Whether it will result in better care for the patients remains to be seen.

## 1. Introduction

The process of developing a certification and grading was inherently challenging. In order to gain an understanding of the subject matter, it is first necessary to provide a brief overview of the historical and developmental context of SCI care in Germany in the period following the Second World War, as well as relevant aspects of the German healthcare system and reimbursement requirements. The following Section will provide a contextual framework for understanding the process described.

### 1.1. German Healthcare

The German healthcare system is divided into two distinct sections: inpatient care and ambulatory medicine. Inpatient care is provided by hospitals that are either run by the local community, the state (public sector: 539 hospitals; 8.3 M cases), non-profit organizations (590 hospitals; 5.8 M cases), or by private institutions (750 hospitals; 3.1 M cases) [1]. Non-profit organizations include social partnership institutions (e.g., church organizations) and hospitals that are funded by workers’ compensation organizations (also known as “employer’s liability insurance associations”). The majority of private companies operate a number of facilities, with some even listed on the stock exchange [2].

The provision of ambulatory care is primarily the responsibility of duly authorized physicians. This authorization is typically associated with a specific individual and qualification (e.g., board-certified orthopedic surgeon, neurologist). Hospitals are granted only limited access to ambulatory care. No authorization is necessary to treat private patients.

The funding of reimbursement is derived from a number of sources, including the compulsory health insurance of every employee (with the exception of civil servants and soldiers, which are covered by federal funding), private insurance, employer’s liability insurance, and social welfare.

Access to non-private reimbursement is subject to stringent regulatory oversight. Hospitals are required to obtain authorization from a central regulatory committee for the assignment of beds. Additionally, authorization must be obtained for ambulatory care services. The annual budget of the German healthcare system is 500 billion euros. The cost of care for paraplegic patients is estimated to range from 350 to 500 million euros [2].

### 1.2. History of the Care for SCI Patients in Germany

The aftermath of World War II saw a significant number of spinal cord injuries among soldiers and civilians, necessitating specialized care and rehabilitation. A mere handful of specialized units existed after the war, with some of these units having commenced operations prior to the conflict. Examples of such units include those in Bochum, Heidelberg, and Bayreuth [3]. Subsequently, other facilities were established, the majority of which were based on the comprehensive care model pioneered by Sir Ludwig Guttmann [4].

The facilities in question were provided either by public entities or workers’ compensation organizations. The advent of privately funded hospitals in the market occurred subsequent to 1980. Today, 10 out of 27 SCI units are privately funded.

The care of members of the German military forces is a responsibility shared by all hospitals, with a particular focus on those operated by workers’ compensation organizations.

It is noteworthy that in Germany, it is not customary for a physiatrist to oversee the operations of a spinal cord injury unit. The majority of these units are managed by orthopedic surgeons, with only a small number overseen by neurologists. The larger units (70–150 beds) provide comprehensive care for individuals with SCI, encompassing a range of services from spine surgery and acute rehabilitation to long-term outpatient care and complete management of complications. These units offer specialized care for neuro-urologic conditions, including surgical procedures, pressure sore management (including flap surgery), and treatment for pain and spasticity. These units have their own intensive care beds and offer weaning facilities. Additionally, there are other units that solely provide conservative management and are exclusively oriented toward rehabilitation.

### 1.3. Reimbursement

It is important to note that, under the German health system, no certification from any professional society is required in order to receive reimbursement. Reimbursement is and was regulated through social legislation (SGB V, [5]). Prior to 2004, hospitals were remunerated on a per-treatment-day basis, with the remuneration amount negotiated with the regulators on an individual basis for each hospital. In determining these remuneration amounts, the regulators took into account the structure and performance of each hospital. This changed with the advent of the DRG (diagnosis-related group) System. Since then, remuneration became more uniform, following the norms set by DIMDI (Deutsches Institut für medizinische Dokumentation und Information; German Institute for Medical Documentation and Information [6]), which is responsible for the coding of medical conditions, and InEK (Institut für das Entgeltsystem im Krankenhaus; Institute for the Hospital Remuneration System). The latter is responsible for the definition of DRGs [7], which determine how much the hospital can charge for the treatment. It also sets the requirements and the extent of medical care needed to be able to claim reimbursement under a specific DRG. In the case of spinal cord injury, these regulations do not include definite structural requirements for hospitals. There is no accreditation system comparable to the CARF (United States Commission on Accreditation of Rehabilitation Facilities). The fulfillment of the requirements for reimbursement is subject to the regulatory oversight of the Medical Service in the Healthcare Sector (Medizinischer Dienst im Gesundheitswesen), a public corporation operating under the legal supervision of the Federal Ministry of Health.

### 1.4. History of DMGP as a Multidisciplinary Society and a Representation of Interests for SCI Centers

Organized contact between the units was minimal until 1984, when the medical directors of select SCI centers convened to establish an informal organization dedicated to unifying efforts in the treatment and rehabilitation of SCI patients across German-speaking countries (DMGP). Almost from the beginning, the society emphasized a multidisciplinary approach, integrating medical, therapeutic, and social aspects of care. The DMGP prioritized research in SCI and facilitated education and training for healthcare professionals, especially by organizing regular conferences. Later, the society developed and disseminated guidelines and protocols to standardize treatment and rehabilitation practices [8]. The DMPG maintained a website with a listing of the SCI centers, the purpose of which was to guide primary care facilities in the transfer of patients and to inform patients and relatives about the nearest possible treatment facility [9]. The facilities in question submitted applications to the board of DMGP. The board granted the applications in all cases without having a clear definition of an SCI center as a basis for this decision.

The list and the process were not the subject of much attention until the reimbursement policies became financially advantageous for non-specialized neuro rehabilitation centers to offer SCI care as an additional service. In many instances, these facilities lacked the necessary expertise in SCI treatment, yet they posed a significant financial threat to the reimbursement of established centers. Consequently, a growing number of objections from established centers were raised, citing that the recently added member of the list did not offer a service that was deemed to be of an adequate nature. In this situation (2015), the DMGP decided to establish criteria for SCI centers and started the development of a grading and certification system.

A total of twenty-seven units specializing in the treatment of spinal cord injuries were affiliated with DMGP. There were eight SCI units that were part of professional association clinics, of which three belonged to a university hospital, seven were publicly funded, and twelve were privately funded. The structural configuration of the units exhibited considerable diversity, with some units affiliated with maximum care providers offering 24 h, seven-day-a-week polytrauma services. Conversely, there are also rehabilitation facilities that lack an intensive care unit but nevertheless provide an excellent service for patients.

It was mandatory that the certification did not go against reimbursement requirements and during the certification process, especially when it came to requirements for physical therapy (PT), occupational therapy (OT), speech therapy, sports therapy, social services, and psychologists. The DMGP initiated the certification and grading process for SCI centers with the objective of maintaining treatment standards, safeguarding affiliated hospitals that adhere to these standards, and enabling the professional community to defend the existing standards against attempts to dilute them for financial reasons. There were no SCI units with a preexisting certification, and the DMGP had no means of enforcing participation in the certification process other than by persuading its members that the certification would be a valuable tool for maintaining and improving the quality of care.

## 2. Methods

The DMGP engaged the services of a team comprising three individuals (K. Rehan, O. Marcus, and B. Domurath) to undertake this assignment. The timeline for the development and implementation of the criteria and requirements for the certification and grading process is presented in Table 1. The group started by defining the scope of the task. From the outset, it was decided that certification should only relate to general structural characteristics; success criteria, such as the functional development of the patients treated, were not included. One of the reasons for this is that only a few centers (e.g., those participating in the European Multicenter Study about Spinal Cord Injury network) collect standardized follow-up data at all.

Literature review has been a part of the entire process. A database search (PubMed) was used to identify work that described SCI care, focusing on publications on the situation in the (German-speaking) European context [10,11,12,13,14,15,16,17,18]. The relevant publications of payers, local committees, and oversight bodies were identified through their respective websites. They were also evaluated [5,6,7,8,10,18,19]. In addition, the available results of similar working groups, especially from Switzerland [11,20], were evaluated.

In addition, a questionnaire was sent to all centers listed to date, asking for information about the personnel and equipment available, as well as the treatment modalities offered. The recipients were the heads of the respective facilities, who are represented in the DMGP’s “Doctors’ Working Group”.

Based on this, the areas of intervention listed in Table 2 were created, divided into the areas of basic care, special care for ventilated patients, self-help training, social-medical support, and complication management.

The fundamental prerequisite for all facilities was the provision of sufficient rehabilitative (conservative) care. Providing acute care only—even with a special focus on SCI, e.g., spine surgery or neuro-urologic care—was not considered to be in line with the accepted premise of “comprehensive care” as described by Guttman [4].

In consideration of the significant supplementary workload entailed by the provision of surgical and intensive medical care, a decision was taken to categorize the facilities into three distinct groups (see Table 3).

In order to gain an overview of the existing personal and material equipment of the centers, a questionnaire was developed and distributed to all SCI centers listed.

In light of the findings of this review, a final questionnaire was constructed as a foundation for a self-reported audit within the context of the certification and grading process. The document was approved by the DMGP board in the spring of 2020 and all represented units through their medical directors.

In the fall of 2020, the units were asked to apply for the inclusion into the future list of SCI centers by the end of 2020, providing completed, self-reported, questionnaires.

## 3. Results

Nineteen of twenty-seven units responded in time, reporting the details of staffing and structure. The results derived from the returned questionnaire were subjected to analysis and discussion. The responding units were twelve units offering surgery and seven offering conservative care only. Some of the centers declined to participate in the study, mostly due to lack of approval by the respective hospital administration.

The unit sizes exhibited considerable variation. Furthermore, it became evident that not all hospitals with the capacity to provide surgical care are admitting patients on a 24 h/seven-day-a-week basis, nor do they offer emergency surgery. In light of these findings, it was determined that the category “capable of surgery” required further delineation into units with and without available polytrauma treatment.

Units offering conservative treatment only tended to be smaller and had fewer medical doctors and nurses per bed. The numbers recorded for the spectrum of therapists were not comparable. Even the numbers for physiotherapists and occupational therapists varied widely. According to the information given in the comments of the questionnaire, the tasks performed varies considerably.

For example, occupational therapists, physiotherapists, or sports therapists may be responsible for selecting an appropriate wheelchair and conducting the requisite training. In some centers, occupational therapists (OTs) perform speech therapy and participate in the weaning and decannulation of respirator-dependent patients, whereas this is otherwise the responsibility of speech therapists. It was thus resolved to aggregate the figures pertaining to the PT and OT personnel in order to obtain a reference number that would serve to quantify the necessity for therapists.

All results are summarized in Table 4.

Based on the results of the review and the literature, the task force drafted a final list of requirements for the acceptance as an SCI center and suggested a final categorization.

There was a consensus that a minimum of 100 patients per year should be treated in the facility, and that the unit should be separated from other hospital facilities.

In addition, basic structural requirements have been established for the units. They request the provision of wheelchair-accessible patient rooms equipped with supplementary devices tailored to the degree of paralysis (e.g., communication systems), wheelchair-accessible sanitary facilities and equipment, and separate therapy rooms with unrestricted access for wheelchair users.

### Categorization of SCI Units

Level 1 comprises units that are equipped with the capacity for immediate post-trauma surgical and conservative care.

The Level 1 category was further divided into two subcategories: Level 1A, comprising units equipped to provide polytrauma treatment on a 24/7 basis, and Level 1B, comprising units lacking the capacity to provide polytrauma treatment.

Units that provided exclusively conservative therapy were classified as Level 2. Once more, a distinction was drawn between units equipped with intensive care facilities (Level 2A) and those lacking such resources (Level 2B). These characteristics are displayed in Table 5.

These results were again summarized and discussed with the board and presented at the annual meeting of the DMGP. It was decided to proceed with the certification process utilizing the suggested templates. In spring 2020, the 27 units received the request to apply for certification in order to remain on the list of SCI centers maintained on the webpage of DMGP. They had to respond to the questionnaire. A total of 25 out of 27 units responded and sent in completed questionnaires.

These self-reported questionnaires were subsequently evaluated by the certification committee of DMGP, resulting in an attribution to a specific level (Table 6).

In the subsequent period, one unit that had not previously participated also submitted an application, and a single newly established center submitted an application and was subsequently certified.

## 4. Discussion

As anticipated, the services provided and the available structures exhibited considerable variation among the SCI centers, which are represented in the DMGP as their local professional association. The range of services offered extended from a 120-bed unit that was part of a supraregional trauma center, which was equipped to handle any emergency 24 h a day, seven days a week, to a rehabilitation-only unit with 30 beds. Both facilities have a long history of servicing patients with spinal cord injuries (SCIs) and are integral parts of the group of specialized care providers for SCIs. The objective was to treat all parties equitably during the certification and categorization process while establishing a reliable basis for evaluating the scope of the respective unit.

Given the observed variety, it became evident that only a limited number of characteristics were consistently present. All units had separate wards and therapy rooms that were fully wheelchair accessible and dedicated to the treatment of SCI patients. Moreover, all units were equipped with the necessary resources for urodynamic measurement. However, some units lacked an in-house urologist to conduct these examinations, instead relying on external consultants.

The results also demonstrate the necessity for differentiation between units that provide surgical services and those that do not. Surgical units are typically larger in size and require a greater number of nurses per bed, although the ratio of therapists may be slightly lower.

The capacity to provide respiratory care tailored to the needs of each patient was widely accepted as a hallmark of more complex care [21]. It became evident that a differentiation of surgical units was necessary, an outcome that was not foreseen. The capacity to provide optimal care on a 24/7 basis necessitates a considerably more substantial investment of resources than the maintenance of a surgical theater, which may be utilized a few times per week.

The resulting stratification into two levels with two subcategories each permitted a transparent categorization process. The focus on the differentiation of SCI units from other facilities that provide only some facets of SCI-specific care or mix SCI patients into a greater “neurologic disorder clientele” was successful. This was achieved by focusing on simple yet essential requirements.

Although it commenced as an enterprise based on scientific principles, it became evident that regulatory stipulations, stipulations established by principal stakeholders, including health insurance organizations and the historical evolution of SCI care, would play a pivotal role in this undertaking [5,7,8,10,17,18]. Given the involvement of long-standing member hospitals of the DMGP, it was imperative to ensure that no party was marginalized.

A major obstacle that was evident reviewing the literature is a paucity of evidence regarding the efficacy of individual interventions [15]. It is evident that physiotherapy or occupational therapy are necessary components of SCI care [13]. Nevertheless, it remains uncertain who is best positioned to ascertain the requisite extent and duration of these therapeutic interventions, particularly given the fact that therapists are tasked with responsibilities that are not always clearly delineated, and which may vary from one unit to another. The same is true with regard to nursing. While work conducted in the European context or for other areas of specialist care was helpful [11,12,14,16,22], it could not be used to delineate clear-cut requirements, such as those pertaining to staffing.

In the meantime, other initiatives have been undertaken, and it is now possible to compare the requirements of DMGP certification with the results of International Spinal Cord Injury Survey (INSCI) [23], the requirements formulated by WRA (World Rehabilitation Association) [24], and the Spinal Cord Injury Service Module Project chaired by Middelton [25]. The toolkit has been endorsed by International Spinal Cord Society (ISCOS), the international professional society for spinal cord care.

It is recommended that the findings of this work be incorporated into future iterations of the DMGP framework for SCI units.

Moreover, it will be necessary to provide statements regarding the inclusion of novel therapeutic modalities, such as those enhanced by robotic devices [26]. Will it be necessary to offer robotic enhanced therapy in order to be accepted as an SCI center?

It can be considered a significant achievement that nearly all member units were able to come together in support of this initiative and accept the proposed certification rating. The decision of the recently inaugurated facility to pursue certification in accordance with the stipulated prerequisites, without any implications for remuneration or the necessity to commence operations, is a testament to its commitment to excellence.

It is also unclear whether the system of comprehensive care will remain viable, given the significant decline in interest among spine surgeons in providing post-surgical care for patients with spinal cord injuries.

This is particularly pertinent given that the certification does not currently have any implications for reimbursement. It remains to be seen whether this work will be adopted by regulators and financial stakeholders. Nevertheless, the majority of contemporary certifications originated in a comparable manner and were ultimately mandated for implementation in medical care domains such as stroke care.

The certification process will be subjected to a review in light of the fact that the initial recertification is scheduled to take place next year. Henceforth, all audits pertaining to new applications and those conducted five years after the certification issue will be on-site audits. This represents a departure from the previous practice of conducting exclusively self-reported audits. Ultimately, the DMGP aims to collaborate with other European professional societies to establish a unified vision for the role of an SCI center in our region.

## 5. Conclusions

The process of grading and certifying spinal cord injury treatment centers is a challenging and time-consuming undertaking. In addition to the academic recommendations, it is imperative to consider the legal prerequisites and remuneration requirements.

Nevertheless, all established centers in Germany were amenable to participating in the process. It remains to be seen whether the desired outcome of maintaining current treatment standards will be achieved.

## Figures and Tables

**Table 1 healthcare-12-02529-t001:** Timeline.

Timeline	Actions of DMGP Board	Actions of Task Force
June 2014 (annual meeting)	Decision to develop an accreditation for SCI units Board of DMGP	
	Commissioning of a task force K. Rehan; O. Marcus; B. Domurath	Stratification of the process defining the scope of the task description of the problems in creating general characteristics for SCI units literature review
2015–2016		Multiple meetings of the task force
2016	Presentation of preliminary results to the board of DMGP and approval to continue	
2016–2017		Questionnaire sent to all SCI units listed on the DMGP website review of results (24/27 centers responded) definition of categories for SCI Units description of specific characteristics for SCI Units
2017	Review of the proposal and by the board and the representatives of the and approval to continue	Discussion and modification of requirements
2017–2019	Decision to implement certification process based on finalized recommendations.	Discussion and modification of requirements
Spring 2020	Start of the certification process by mailing self-reported questionnaire to member units	Finalizing the forms for self-reported audit of the characteristics of the SCI unit
Fall 2020	Issuance of certificates (valid for 5 years)	Review of returned questionnaires, classification of centers

**Table 2 healthcare-12-02529-t002:** Therapeutic interventions.

**Basic therapy:** Physiotherapy, occupational therapy, or sports therapy; predominantly as individual therapy with paraplegic-specific content Sports therapy with wheelchair training Training of activities of daily living Mobility training Testing and adaptation of aids Wheelchair adaptation (various models, including shower commode chairs)
**Advanced therapy for ventilated patients:** Therapy and prescription of aids for various issues relating to continuous ventilation, including weaning (permanent/partial) from the ventilator Check-ups and treatment of complications during continuous ventilation
**Training to establish selfcare for vital care:** Learning self-catheterization under the guidance of nursing staff and testing suitable catheter systems, considering the level of paralysis Setting up adequate bowel management under the guidance of nursing staff Learning adequate positioning under the guidance of nursing/therapy Prescribing/testing positioning aids (including special mattresses) Guiding relatives and instructing professional nursing staff in the special requirements of paraplegics
**Socio-medical support:** Helping to submit the necessary applications to authorities and funding agencies Organization of professional reintegration Sexual counseling Patient and family education about SCI and its complications
**Therapy of complications** pressure ulcers, urological complications, urological complications, neuropathic pain, spasticity, orthopedic and trauma related complications, vegetative dysreflexia Therapy to compensate for chronic, SCI-specific functional deficits

**Table 3 healthcare-12-02529-t003:** Initial proposal to categorize spinal cord injury units. It was imperative that the services in question be provided without the transfer of the patient into other hospitals within the premises of the unit by the staff of the unit. (Cooperation, e.g., with a plastic surgery department in a nearby hospital, did not qualify as “provision of surgery”).

Category	Conservative Care/Rehabilitation	Intensive Care Unit/Treatment of Respirator Depended Patients	Provision of Surgery
Category I	yes	yes	yes
Category II	yes	yes	no
Category III	yes	no	no

**Table 4 healthcare-12-02529-t004:** Results of questionnaire sent to affiliated hospital.

Structure	Unit Separated from Other Parts of the Hospital; Dedicated Therapy Rooms; Wheelchair Accessibility of the Whole Unit	19/19
Treatment capacity Surgery provided	Number of own beds	31–120 (mean 66.5)
	SCI patients per year (inpatient treatment)	300–1100/year
Treatment capacity Conservative treatment only	Number of own beds	27–45 (mean 32.4)
	SCI patients per year (inpatient treatment)	220–400/year
Staffing	Number of doctors	4–22
	Specialties Doctors	Orthopedic surgery, neurology, physiatry, internal medicine, urology
	Number of nursing staff (surgery available)	39–161 0.88–1.3/per bed
	Number of nursing staff (conservative treatment only)	24–40 0.45–0.89/per bed
	In-house urologist available	18/19
	Physiotherapy/Occupational therapy (surgery available)	0.26/bed (0.12–0.44)
	Physiotherapy/Occupational therapy (Conservative treatment only)	0.31/bed (0.22–0.4)
	Speech therapy	1.72 *
	Psychologist	1.49 *
	Social services	0.85 *
	Sports therapy	1.62 *
Operative equipment	X-ray, Computed tomography (CT), Magnetresonance tomography (MRT)	15/19 on site, 4/19 CT/MRT available on demand with transport required
	Sonography	19/19
	Laboratory	19/19
	Bacteriology laboratory	19/19 (if not on site, available on a demand basis)
	Ventilation treatment stations for permanently ventilated patients	12/19
	Urodynamics	19/19
	Electrophysiology	15/19
Participation and membership in the professional association DMGP	Medical director active in DMGP	19/19
	Participation in working groups of the DMGP	19/19

* = for the whole unit.

**Table 5 healthcare-12-02529-t005:** Categorization of SCI centers.

Level 1 Operative and Conservative Treatment of Acute and Chronic Paraplegia (Traumatic and Non-Traumatic Origin) and Treatment of SCI Specific Complications		Level 2 Conservative Treatment of Acute and Chronic Paraplegia as Well as Treatment of Traumatic and Non-Traumatic Causes of SCI Specific Complications		
Level 1a: Including Polytrauma—Treatment 24/7	Level 1b Excluding Polytrauma—Treatment	Level 2a: ICU or Treatment of Respirator Dependent Patients Possible	Level 2b: Treatment of Respirator Dependent Patients Not Possible	
yes	yes	yes	yes	Inpatient treatment of at least 100 pat. with SCI/year last 5 years
yes	Not mandatory	Not mandatory	no	Intensive care Unit
yes	yes	yes	no	Treatment possibility for ventilated patients
yes	yes	no	no	Operating rooms
yes	yes	yes	yes	Laboratory diagnostics, including microbiology
yes	yes	yes	yes	Sonography
yes	yes	Accessible (not necessarily inhouse)	Accessible (not necessarily inhouse)	X-ray/CT
yes	yes	no	no	Diagnostic and interventional radiology with large-scale equipment (Angiography, MR, multifunctional workstation; CT)
yes	yes	yes	yes	Video-urodynamic measuring capability
Not necessary but desirable				
yes	yes	no	no	Specialized outpatient and neuro-urological aftercare
yes	yes	yes	yes	Neurography/evoked potentials

**Table 6 healthcare-12-02529-t006:** Results of the certification and categorization.

Level	Number of Units (25 of 27)
1a	13
1b	6
2a	5
2b	1

## Data Availability

The data collected on individual SCI units participating is confidential and not available.

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
