# Peer review of "The Development of a Certification and Grading Procedure for German SCI Units"

_healthcare, 2024, doi:10.3390/healthcare12242529_

Round 1
Reviewer 1 Report
Comments and Suggestions for Authors
Reviewer Comments
First of all, this is really interesting information for many across the globe to understand German SCI care and learn about your comprehensive process to standardize and document care across your nation. Thank you for doing this work. I think it is important and very worth publishing.
OVERALL: The below suggestions are made with the intention of strengthening the manuscript for publication. Please consider adding more context on German healthcare system and reimbursement processes in the background, and more intentionally set the stage for “we realized we needed to find out what the 27 SCI Centers provided, standardize ideal facility recommendations, and establish facility levels.” Something to this effect at the end of the Background. Then, please articulate your process from beginning to end in the Methods section, with a Figure timeline for visual communication. Please present only data in the Results section. Please put any interpretation of results, recommendations, and conclusions in the Discussion section.
To help with relevance to a wider/international audience, the following ADDITIONAL CONTEXT AND BACKGROUND could enhance understanding and impact:
* How do the identified SCI centers relate to the German healthcare system today as well as during/after WWII? Are they private, national/government or part of a national healthcare system? Does Germany provide healthcare to active duty military personnel and veterans separate from private or national health care system? I think this context is important (as would be a few more references to support it). Thank you for considering.
* Reimbursement policies are brought in at line 59...suggest also providing payor system context at the outset along with private/government/separate military healthcare context, and specify the source/organization doing the reimbursement.
* line 66: please provide/cite the DMPG website with the SCI center listings.
line 77: "Each unit was sponsored by a different entity." This is too vague for an international audience. The context information suggested above could assist with being more specific rather than using "entity."
* line 93 is the first mention of "German health system," so please put this into more detailed context at the beginning, to assist the international audience. Thank you for considering.
* line 93-94: "certification from any professional society": Please define further perhaps by listing/citing professional societies that do provide certification. My bias is United States' Commission on Accreditation of Rehabilitation Facilities (CARF) which does have SCI accreditation. Did any of the 27 facilities have an accreditation like this? Please state X/27 and what accreditation organizations they worked with, if any. The X/27 (especially if 0), would be helpful data to set the stage for the need for the certification effort.
* at the end of the BACKGROUND section, please have a summary paragraph identifying the specific need, the main objectives, and what is the hoped impact.
* please be mindful to put all of the process, from “realizing the need to do this” all the way to finishing the process, in the Methods section. Please try to keep the process out of the other sections. Please try to keep only data in the results section. Please try to keep all of these items in the Discussion only: interpretation of the data, recommendations, and conclusions (how did the objectives get met, and what is the impact for the international audience).
OTHER SUGGESTIONS
* line 69-70: "in a decision that was arguably arbitrary..." this language seems more appropriate for an opinion or editorial article. Suggest taking out language that takes a more judgmental tone and reporting in more dispassionate, factual manner that allows reader to come to own conclusions. The entire paragraph lines 66-73, suggest presenting in more factual manner...here is the website, applications were submitted, XX/XX were granted (presenting actual numbers would be most helpful). Perhaps consider "There was a perception of lack of standardization" or something like that, which propelled the establishment of criteria.
* lines 85-86: not sure about the "dosage" evidence; however, please consider framing with brief summary with 2-3 references of literature on functional rehabilitation gains over time and rehabilitation efficiency for paraplegia and tetraplegia. And please comment on how the 27 Centers compared to providing the multidisciplinary rehabilitation care--and do any of the 27 Centers measure rehabilitation functional gains and length of stay?
* lines 99-120: This summary of criteria could be improved in a couple of ways. First, the citations/references that are not in English--if at all possible, please provide alternatives in English. Second, the information could be streamlined...instead of lines 99-120, assuming this is a description of the citations/references used to create Table 1, it might work better to get rid of lines 99-120 (which seems like a list of references) in favor of a shorter description and then place reference/citations within Table 1. We did something similar for a Table in a recent publication and it worked well to streamline the information and shorten the narrative, making room for other important information.
* line 123 "the group than began" change "than" to "then"...however suggest this sentence could work better, something like, "After establishing the list of therapeutic interventions (Table 1), three facility categories were established." Also please state the rationale for developing the facility categories.
* line 129: #3: Consider aligning with #1 and #2 in language, maybe something like "SCI centers that offer conservative therapies and do not have in-house intensive care unit." This makes it more clear, assuming the #3 facility does not have the ICU. Also a Table could be added with a row for each facility level and "X" in columns of conservative therapies, surgical treatments, and ICU, might be a more streamlined way to communicate the information.
* For Table 2, could data be presented…if not specific to each of the 27 centers, perhaps the average #s of staff, XX/27 for having various equipment? Then suggest connecting these data points to the recommendations provided in the results section. Please consider presenting data in Table 2 and then summarizing the data results in the results section, and then save the comments like “demonstrated that the management of SCI rehabilitation can be highly variable yet still successful” in the Discussion section. The bulk of the Results section is discussion, so please move it and reframe the Results section as a presentation of data only—saving data interpretation and recommendations for Discussion section.
* Table 3 is confusing. The title is "Equipment," but lines 142-145 describe it as "basic requirements." Please present data for Table 3, similar to what is suggested for presenting data for Table 2.
* Table 4, suggest something similar to recommendation above for line 129, to create a new table. Table 4 could be reformatted so that there is a row for each characteristic (e.g., inpt tx of at least 50/yr), and then a column for Level 1a and 2a. Then place an X in the box when the Level has that characteristic. This conveys the information faster, more visually, than having to read each list and decipher what is common/different.
* Table 5, same suggestion as for Table 4.
* lines 168-171, 176-188: this is a timeline—suggest making a Figure with a timeline for the key events of the process, some of which is described in all sections—intro, method, results, discussion. This would assist in helping the reader frame the process and streamline the narrative. Suggest that the entire timeline of realizing the establishment of certification criteria was needed, all the way to setting up for future audits…this should be a timeline presented (preferably with a Figure) in the Methods.
* lines 172-175: this is Methods section information, please move.
DISCUSSION SECTION
* create a first paragraph that summarizes the data presented in results
* create two separate paragraphs for interpreting the data and recommendations moving forward.
* create a final paragraph with your main conclusions…was the objective (stated in the last sentence of the Background section) achieved? What is the impact on German SCI care and Germans with SCI moving forward? How is this important to other nations’ SCI providers?
EDITS:
* line 43 "Almost from the beginning,..." needs a comma after "beginning"
* lines 56-58: suggest putting the single sentence into above paragraph
* lines 74-75, bring single sentence into prior paragraph
* line 74: recommend changing "criterions" to "criteria"
* line 84: "...and services like that." Too informal. Suggest listing all or most of the disciplines.
*l line 99-100 recommend no single-sentence paragraphs, suggest to put with paragraph below
Author Response
The comments to the review are submitted in the requested form via the attached rebuttal letter.

Reviewer 2 Report
Comments and Suggestions for Authors
The manuscript on the development of a certification and categorization system for the treatment of spinal cord injury (SCI) patients addresses a very significant topic. However, it lacks scientific analysis, and therefore, overall improvement is necessary.
Comment 1: Supplement the Introduction and Discussion sections by citing more references.
Comment 2: If you conducted a literature review and a questionnaire as part of the process, provide detailed descriptions of these procedures and methods.
Comment 3: Present the results obtained through scientific procedures.
Author Response
Thank you for your comments. The paper was rewritten in most parts to accommodate the advice give by you and reviewer 2.
Comment 1: Supplement the Introduction and Discussion sections by citing more references.
References were added
Comment 2: If you conducted a literature review and a questionnaire as part of the process, provide detailed descriptions of these procedures and methods.
The literature review was conducted by a pubmed search, the literature used had a focus on the situation in the local European area (Germany/Austria/Switzerland). The development of the questionnaire is described in the text.
Comment 3: Present the results obtained through scientific procedures.
The results are now provided (Table 4)
Round 2
Reviewer 2 Report
Comments and Suggestions for Authors
- The authors have appropriately revised or responded to the reviewers' comments from Round 1.
- However, for better readability, I recommend merging the shorter paragraphs in the Introduction (Lines 98-140) with the preceding or following paragraphs.
- There are duplicate reference numbers in the reference list. Please correct this issue.
Author Response
However, for better readability, I recommend merging the shorter paragraphs in the Introduction (Lines 98-140) with the preceding or following paragraphs.
This has been done
There are duplicate reference numbers in the reference list. Please correct this issue.
This has been corrected.
